# Detecting the Bronze Age Sites by Using CORONA Satellite Photography and UAV Photogrammetry: A Case Study from the Middle of Yangtze River, China

**Qiushi Zou** [1,2,3]

1 School of History, Wuhan University, Wuhan 430072, China; zouqiushi@whu.edu.cn
2 Institute of Yangtze River Civilization and Archaeology, Wuhan University, Wuhan 430072, China
3 Intellectual Computing Laboratory for Cultural Heritage, Wuhan University, Wuhan 430072, China

**Abstract:** The CORONA satellite image preserves the landscape from half a century ago, and has played a great role in landscape archaeology in many regions of the world. In recent years, with the rapid development of UAV (Unmanned Aerial Vehicle) Photogrammetry technology, Archaeologists can easily obtain the digital surface model (DSM)and Digital Ortho Map (DOM) of a site in the fieldwork. In the archaeological survey of bronze age sites in the middle of the Yangtze River project, we combined the UAV photogrammetry results with CORONA satellite photography, which can help us extract the surface landscape feature of the sites. This strategy has shown significant advantages in reconstructing the settlement layout, detecting the unknown linear features (such as walls, moats and canals) of sites and comparing the landscape between different sites.

**Keywords:** CORONA satellite photography; drone; bronze age; Yangtze River





## 1. Introduction

The CORONA satellite is a well-known American satellite. During the operational phase of CORONA between 1960 and 1972, over 800,000 high-resolution images were acquired. In 1996, a large part of the imagery was declassified and made available on the Internet; in 2002, a second part was declassified. During the past 20 years, CORONA imagery became a valuable tool for detecting different types of unknown archaeological features such as Syrian hollow ways or abandoned Iranian pastoral campsites [1,2], Southern England archaeological crop marks, and flood management landscape in the middle Nile Valley in Egypt [3,4]. Since the 1990s, historical aerial and satellite images have been used in Chinese archaeological survey projects to detect archaeological sites and map the layout of ancient settlements, especially in Xinjiang in northwest China, such as the exploration of the Beiting and Gaochang ancient city sites [5,6]. In the past 20 years, CORONA images have been used in archaeological projects in China. With the help of CORONA images, Chinese archaeologists discovered the water control engineering system from 5000 years ago in Liangzhu ancient city and the city sites from 400 years ago in Xinjiang in northwest China [7,8]. However, CORONA imagery lacks spatial reference information and has varying degrees of geometry distortion, and their ground resolution is usually 2–8 m. Obviously, the limitations of the CORONA image also exist in terms of landscape archaeology research. Archaeologists are now regularly incorporating multisource high-resolution remote sense data into their research projects, especially taking advantage of the UAV photogrammetry to obtain the digital surface model (DSM), Digital Ortho Map (DOM) and 3D model of a site with the help of computer [9,10]. DEM and DOM have a precise geographic coordinate system, and their ground resolution is 2–5 cm, which can make up for the weakness of the CORONA image. Therefore, the combination of CORONA image and UAV photogrammetry is an effective way to carry out landscape archaeology.

Since 2020, our research team has conducted archaeological surveys and mapping of more than 30 sites in the middle reaches of the Yangtze River with the help of UAV photogrammetry and CORONA satellite imagery. We found out that the integration of multisource remote sensing data plays a significant role in detecting the detailed characteristics of archaeological sites, recording site landscape information, and restoring the layout of ancient settlements. Nowadays, UAV photogrammetry is widely used in the field of archaeology. In fact, CORONA satellite imagery recorded the surface landscape half a century ago and combining historical satellite imagery with modern UAV photogrammetry is undoubtedly a very efficient way to conduct archaeological investigations. However, it is worth noting that remote-sensing imagery does not solve all archaeological problems, so remote-sensing archaeological investigations must be combined with field archaeological work. Finally, archaeological excavations and exploration were used to confirm the features found in the remote sensing imagery.

## 2. Study Area

The middle of the Yangtze River is located in the south of China, with abundant rainfall and numerous rivers and lakes. The landform is dominated by alluvial plains of rivers and lakes but also includes a small number of hills and mountains. On the alluvial plains, modern villages are mostly spread over mounds. This kind of mound is 2–5 m higher than the surrounding surface, which can effectively resist the flood threat brought by the rainy season. Interestingly, many bronze age settlements in this region are also distributed on some mounds, similar to modern villages. Therefore, during the archaeological investigation, these mounds or hills several meters above the surrounding surface often attract the attention of archaeologists.

Before the 1970s, large-scale infrastructure construction had not been carried out in this region, and the land use pattern was mainly farmland. Many archaeological sites of the prehistory and bronze age have been preserved in the fields. Since the 1970s, large-scale infrastructure construction and urban expansion have led to the disappearance or destruction of many archaeological sites, which has brought great challenges for modern archaeologists to observe the layout of ancient settlements.

The field archaeological work in this region began in the 1950s; however, before the 21st century, the field archaeological survey in this region was mainly conducted by walking on the ground, and almost no modern spatial information technology was used to detect and analyze the sites. In recent years, UAVs have been increasingly used in the investigation of archaeological sites, which enables archaeologists to observe the shape and characteristics of ancient sites from a broader spatial perspective. The results of UAV photogrammetry enable us to measure and analyze the overall layout and detailed characteristics of ancient sites. The innovation of these technical means undoubtedly provides very powerful support for archaeologists to study the ancient site landscape.

The bronze age of China roughly coincided with the Shang and Zhou dynasties in Chinese history (1600BC–256BC). In the Shang and Zhou dynasties, the capital cities were always distributed in the north of China. The Yangtze River valley concerned in this paper is located in the southern territory of the Shang and Zhou dynasties. In the bronze age, in order to plunder and control the copper resources in the middle of the Yangtze River, the Shang and Zhou dynasties continued to expand to this region. Under the above cultural background, a series of settlements of different scales and complex social structures have been formed in the middle of the Yangtze River. Some large and high-grade settlements are often distributed on important traffic routes; they also have rammed earth walls and moats around the walls. These bronze age settlements are important objects for us to study the cultural communication and transportation routes between the north and the south. These settlements are often distributed on soil mounds 2–5 m above the surrounding ground, and some settlements are surrounded by city walls and moats. Therefore, we can detect these spatial morphological features through satellite and UAV images to restore the layout of ancient settlements. Sometimes some unknown site features can be found in

satellite images, which can help archaeologists to reevaluate the scale and grade of the sites. Moreover, by comparing the construction methods of different settlements, archaeologists can also find the diversity of settlement functions in this region so as to better understand the complexity of ancient social structures.

From 2020 to 2022, the Yangtze River Civilization Archaeology Research Institute of Wuhan University conducted field surveys and UAV photogrammetry on the bronze age sites distributed in the middle of the Yangtze River (Figure 1). At the same time, CORONA satellite images covering the site's area were collected. Although these sites were reported in the archaeological reports published in the past, we have successfully reconstructed the layout of some ancient settlements that no longer exist in this project. In addition, we also found previously unknown site features in some sites (such as walls and moats). We also compared the construction methods of different sites and found the diversity of construction methods of Bronze Age sites in this region.

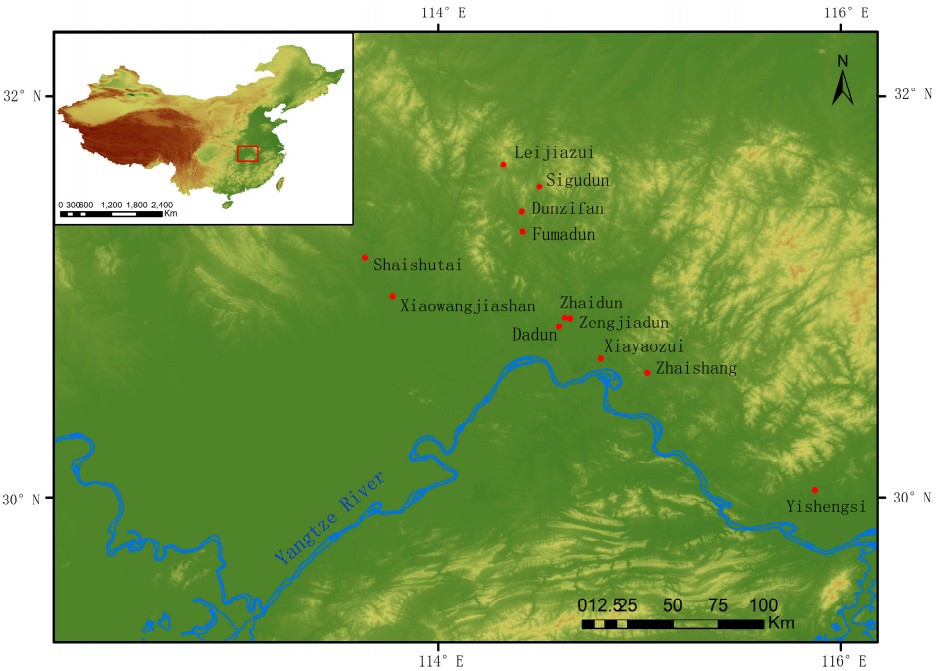

**Figure 1.** Distribution map of bronze age sites in this survey.

### 3. Methodology

The methodology followed in this article is briefly described here. The methodology is made up of four steps starting from the search and download of CORONA images, followed by using UAV photogrammetry technology to produce DSM and DOM. Then the geometric distortion of the CORONA image is corrected based on the DOM. Finally, the interpretation of these remote sensing images and analyzing the landscape of sites are carried out.

Step 1: Acquisition of CORONA imagery

CORONA data are not supplied in digital form but as photographic products. These data are available from the United States Geological Survey Global Land Information System (USGS). The data are inexpensive, easy to obtain, and relatively well documented, and the database of available images can be searched by geographic coordinates and scenes inspected over the web at the USGS website (EarthExplorer (https://www.usgs.gov) accessed on 12 February 2023).

Step 2: Extracting DSM and DOM by UAV photogrammetry

In this project, we use unmanned aerial vehicles (DJI Phamton 4R) to carry out aerial photography operations, which can automatically fly along our planned route and then automatically return. The 3D modeling software we use is Dasearth, which can derive high-resolution DSM and DOM from 3D models.

Step 3: Orthorectification of CORONA imagery

In order to reduce the geometry distortion of CORONA imagery, we select a series of GCPs (Ground Control Points) from the high-resolution DOM (2–5 cm), which are extracted by UAV photogrammetry. Then CORONA imagery is corrected, and obtain WGS84 coordinate system with the help of computer software ArcGIS 10.3.

Step 4: Interpretation of the remote sensing images and analyzing the landscape of sites

With the support of Arcgis 10.3 software, different remote sensing images are overlapped and analyzed. Then we can analyze the dynamic landscape from multisource remote sensing images, which can help us reconstruct the layout of ancient settlements, explore the features of archaeological sites, and compare different settlement construction methods.

## 4. Results

1.    Reconstruction of the layout of ancient settlement

Lvwangcheng is a site located at the southern foot of the Dabie Mountains north of the Yangtze River. From 1979 to 1982, the Hubei Provincial Institute of Cultural Relics and Archaeology carried out two archaeological investigations on the site and found a large number of relics and cultural layers of the Neolithic and Bronze Age [11–13]. Local residents said that the earth wall still existed in this site in the 1960s; however, when archaeologists investigated the site in the 1980s, only about 100 m long rammed earth wall was found in the northwest of the site. The remaining part of the wall has been occupied by modern cities and roads.

In October 2021, we conducted a field survey and UAV photogrammetry of the Lvwangcheng site. At the same time, we acquisitioned the CORONA image covering this area, which was taken in September 1968. In the image processing phase, we used Dasearth to generate 3D models from more than 1000 UAV photos and then derived DSM and DOM. In the CORONA image, we can see that the layout of lvwangcheng is nearly oval and is obviously several meters higher than the surrounding surface (Figure 2). However, in the image taken by UAV in 2021, Lvwangcheng has almost been occupied by modern towns. The northwest of the site is a local high school, the middle and south of the site are modern urban buildings, and the location of the east wall is covered by a modern road. In DEM, we can measure the geometric information of the site. The maximum length of Lvwangcheng is 285 m, the maximum width is 153 m, and this site is 3–6 m higher than the surrounding surface. From the profile map, we can see that there may be a ring of the moat outside the rammed earth wall. However, it is difficult to identify the moat on the modern surface, and further archaeological drilling is needed to confirm it.

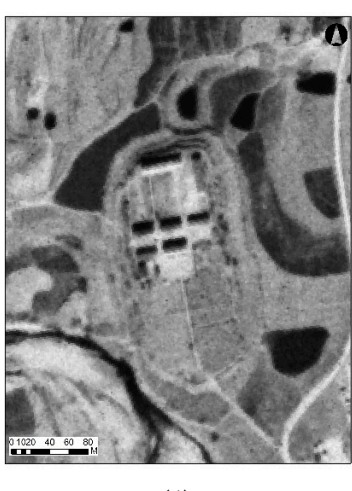
(1)

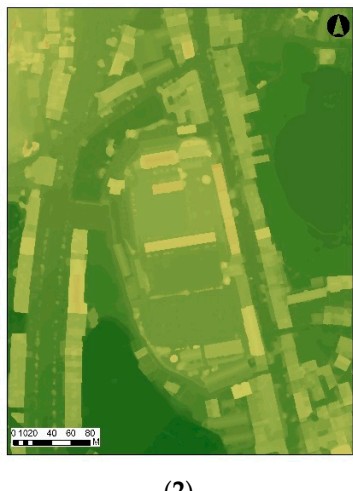
(2)

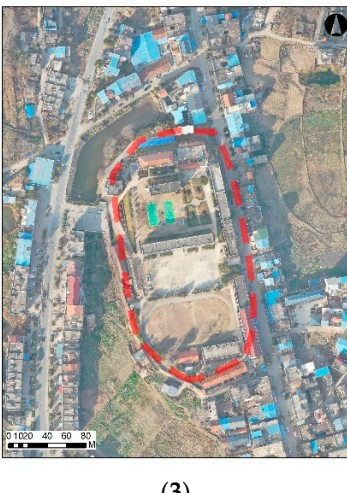
(3)

**Figure 2.** Remote sensing images of the Lvwangcheng site ((**1**) is the CORONA satellite image taken in 1968, (**2**) is the DSM of lvwangcheng, (**3**) is the DOM of Lvwangcheng. The red dotted line in (**3**) shows the scope of the city wall).

2.    Detecting the unknown features of an archaeological site

Yishengsi is located on the alluvial plain north of the Yangtze River. This site is distributed on a mound 5–6 m above the surrounding surface, which is very obvious in the field. In 1996, this site was found during the construction of the highway, which passed through the southern edge of this site. Hubei Provincial Institute of Cultural Relics and Archaeology has carried out an archaeological excavation of this site and excavated relics of the bronze age [14]. However, archaeologists did not carry out a comprehensive investigation of the site at that time.

In April 2022, we carried out a field survey on the site and found that the surrounding area of the site is surrounded by a ring of ditches similar to a moat. Then, we downloaded the CORONA image, which covered the area of this site and used a UAV to conduct the aerial survey. In the CORONA image, we can clearly see a circle of Earth's city wall distributed at the edge of the Yishengsi site, and a circle of the moat is distributed around the city wall. From DEM, we can measure that the maximum length of Yishengsi is 406 m and the maximum width is 218 m. The mound in which Yishengsi is distributed is 5–6 m higher than the surrounding ground surface, the width of the city wall is 10–15 m, and the width of the moat is 20–30 m (Figure 3). It can be seen from the DEM that the north of this mound is significantly higher than the south. For the bronze age site in China, the highland in a city site is probably the distribution area of high-grade buildings. Therefore, it is necessary to carry out archaeological excavation and drilling in the north of the site in the future.

Before this survey, Yishengsi was regarded as a common site of the bronze age. However, we found that there were rammed earth walls and moats in the Yishengsi for the first time, which were not possessed by common sites at the same time. The new discovery of the Yishengsi indicates that this site is a bronze age site with a high social grade in the middle of the Yangtze River. At the same time, this discovery also provides important support for archaeologists to formulate archaeological excavation and drilling plans and also allows us to rethink the social structure of the middle reaches of the Yangtze River.

3.    Document and compare the landscape of different sites

In the survey from 2020 to 2022, we conducted field surveys and unmanned aerial vehicle oblique photogrammetry on more than 30 bronze age sites in the middle of the Yangtze River. UAV photogrammetry can help us quickly and conveniently record the geospatial information of the site, including the longitude and latitude of the site, altitude, geomorphic form, construction method, and the relationship between the site and natural landform, which is a very effective tool in archaeological investigation. When we compare the landscape of different types of sites, we will find interesting information.

On the one hand, in terms of the site area, most of the site area is about 10 thousand square meters(Table 1). This reflects the scale of the most common settlements in the region during the bronze age. We can evaluate the population and the density of the settlements in this region according to the size and spatial distribution of the settlements.

On the other hand, in terms of the type of site, we can divide them into three types [15]. Type A settlements are directly distributed on the natural hills, and the shape is very irregular, such as Daun, Xiayaozui, and Zhaishang (Figure 4(11,12)); Type B settlements are built on the artificial mound, the plane shape is nearly square or oval, such as Zengjiadun, Dunzifan, Fumadun (Figure 4(2–10)); Type C settlement has a regular plane shape, with a circle of the city wall and a moat around the settlement, and the area is also significantly larger than others, such as Yishengsi (Figure 4(1)).

Obviously, compared with directly selecting the natural hill to build a settlement, it will definitely cost more human and social resources to build a mound with a regular shape and even a city wall and moat. This means that settlements with complex construction methods are likely to have higher social levels and undertake complex social functions (Table 1). However, settlements with relatively simple construction methods may exist as common villages. However, in the previous archaeological survey, the microgeomorphology of the settlement has not been recorded in detail and given enough attention by archaeologists.

In fact, it is very helpful for us to study the settlement patterns and social structure of this region by recording and comparing the landscape between different settlements.

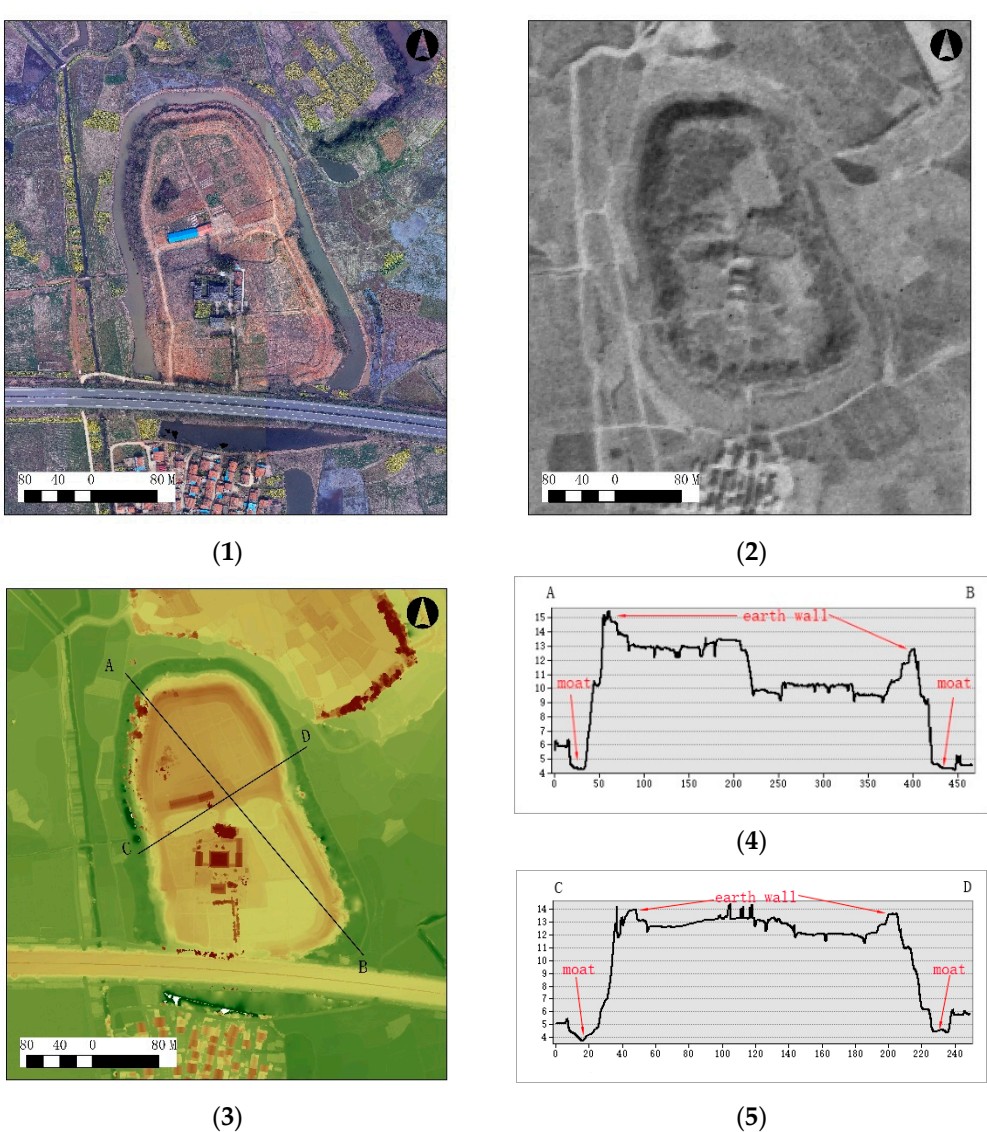

**Figure 3.** Remote sensing images of the Yishengsi site ((**1**) is the DOM of Yishengsi, (**2**) is the CORONA satellite image taken in 1970, (**3**) is the DSM of Yishengsi, (**4**,**5**) is the profile of Yishengsi).

**Table 1.** Sites coordinates and layout information.

| Site | Coordinate | Elevation | Shape | Area (m$^2$) |
|---|---|---|---|---|
| Dunzifan | 114.3253 E, 31.659817 N | 163.5 | Elliptic | 1751 |
| Sigudun | 114.416048 E, 31.423182 N | 64.9 | Circle | 16,726 |
| Leijiazui | 114.420243 E, 31.324574 N | 45.5 | Circle | 2815 |
| Fumadun | 114.503509 E, 31.550313 N | 82.7 | Square | 5623 |
| Yishengsi | 115.8733177 E, 30.03612684 N | 12.9 | Elliptic | 100,527 |

**Table 1.** *Cont.*

| Site | Coordinate | Elevation | Shape | Area (m²) |
|---|---|---|---|---|
| Shaishutai | 113.6375892 E, 31.19509047 N | 41.2 | Square | 7956 |
| Zhaishang | 115.0394661 E, 30.62190842 N | 36.2 | Elliptic | 20,262 |
| Zengjiadun | 114.6564424 E, 30.89109422 N | 50.9 | Rectangular | 9532 |
| Dadun | 114.6010923 E, 30.8495437 N | 34.4 | Elliptic | 9332 |
| Zhaidun | 114.6302426 E, 30.89518199 N | 34.5 | Elliptic | 2289 |
| Xiaowangjiashan | 113.7747788 E, 31.00284662 N | 24.1 | Elliptic | 14,245 |
| Xiayaozui | 114.8081374 E, 30.69486987 N | 27.6 | Rectangular | 50,223 |
| Lvwangcheng | 114.4766255 E, 31.50678997 N | 75.2 | Elliptic | 42,729 |

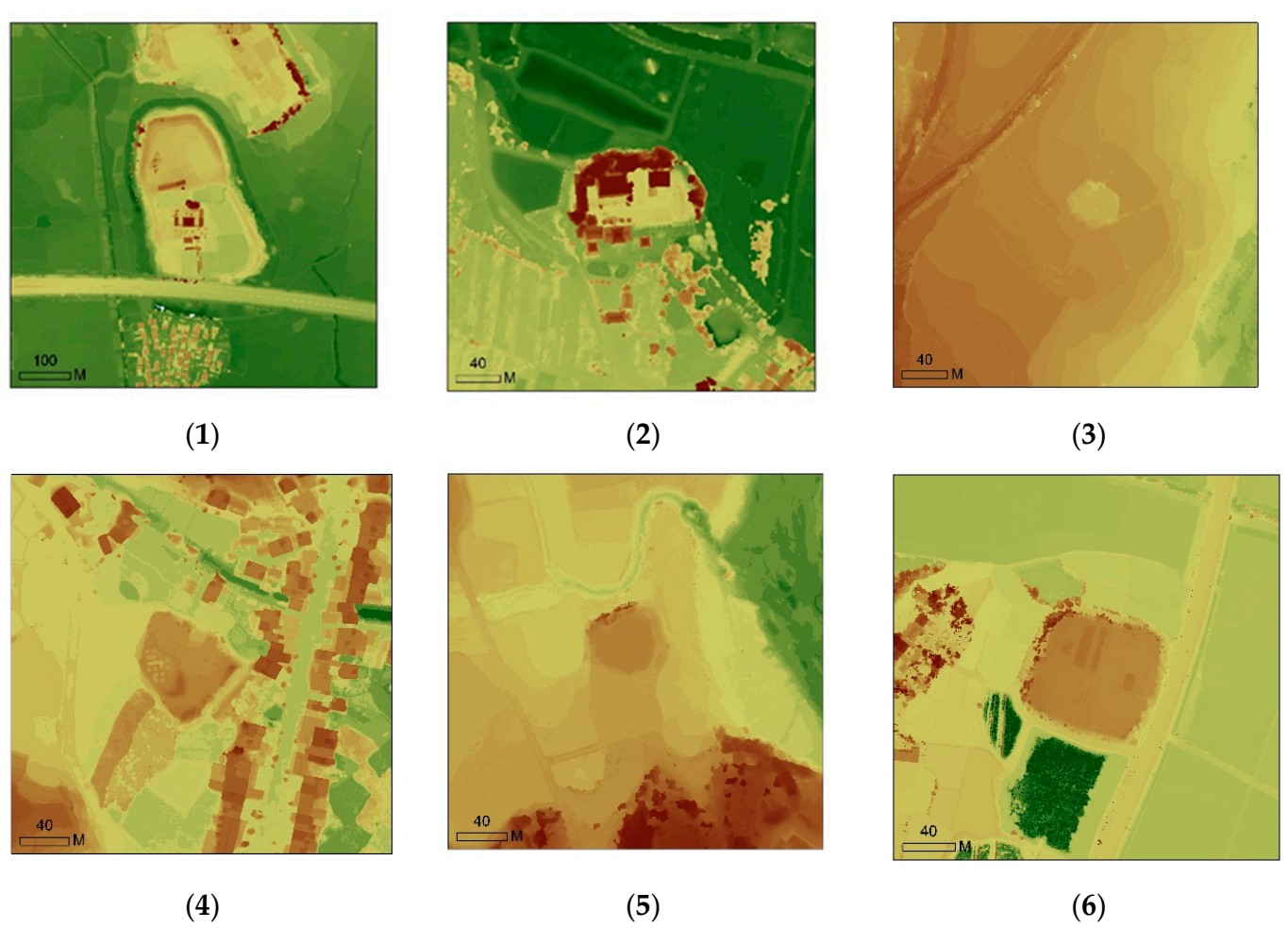

(**1**)　　　　　　　　　　　　(**2**)　　　　　　　　　　　　(**3**)

(**4**)　　　　　　　　　　　　(**5**)　　　　　　　　　　　　(**6**)

**Figure 4.** *Cont.*

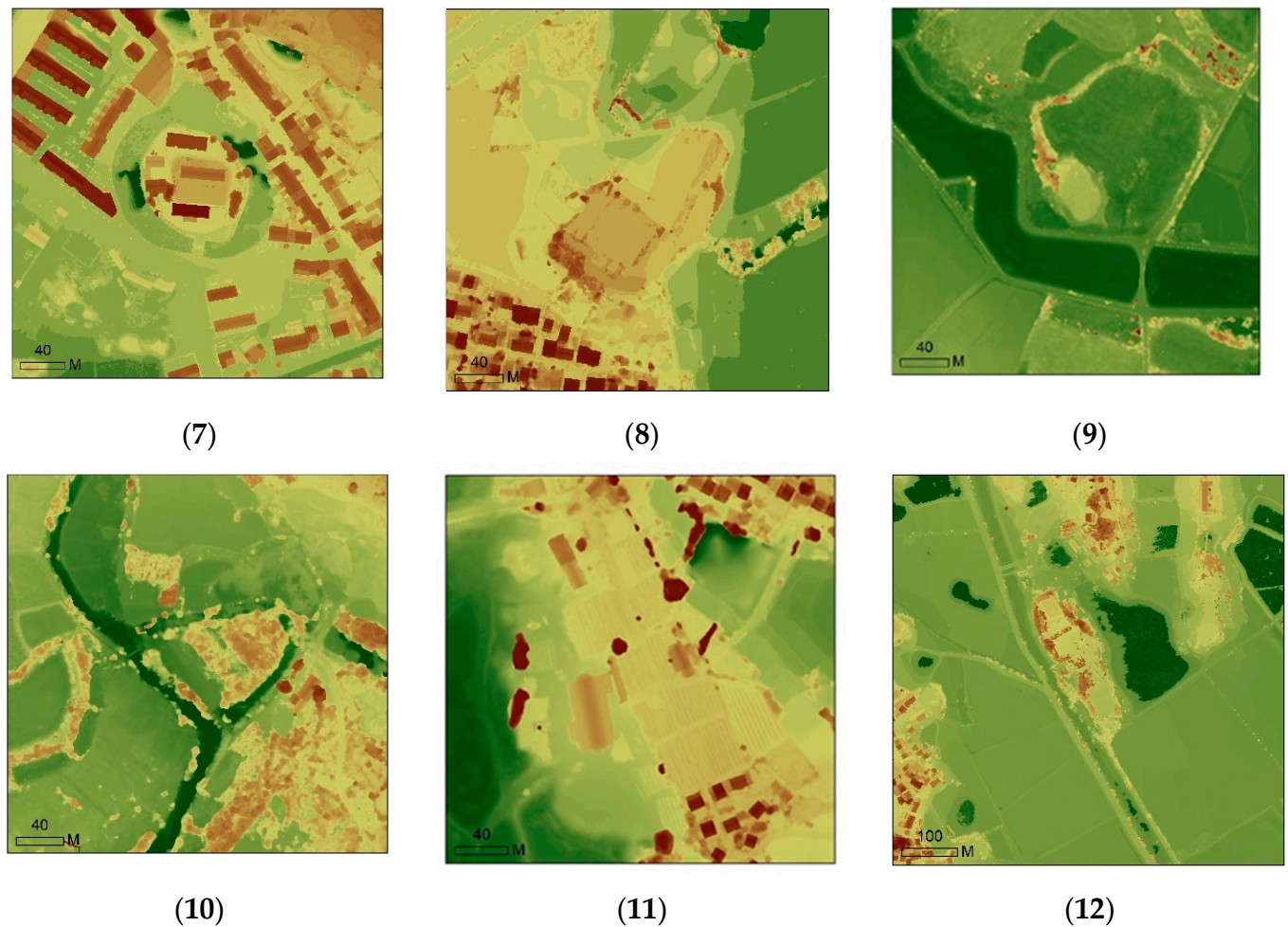

**Figure 4.** Different types of settlement construction methods, (Type A: (**10**) Dadun (**11**) Xiayaozui (**12**) Zhaishang, Type B: (**2**) Zengjiadun (**3**) Dunzifan (**4**) Fumadun (**5**) Leijiazui (**6**) Shaishutai (**7**) Sigudun (**8**) Xiaowangjiashan (**9**) Zhaidun, Type C: (**1**) Yishengsi).

## 5. Discussion

The development of spatial information technology has undoubtedly profoundly changed the ability of archaeologists to detect and record archaeological remains. The middle of the Yangtze River in China belongs to the subtropical monsoon climate zone, with abundant precipitation, a dense modern population, and the surface is dominated by rice farming. Such land use patterns are significantly different from the arid and semi-arid regions in the Near East, Central Asia, and Northwest China. In previous studies, CORONA satellite images have played a significant role in arid and semi-arid regions [16,17] because many prehistoric and bronze age canals, irrigation systems and road systems have been preserved under dry and rainless climate conditions. However, in the middle of the Yangtze River, these relics are difficult to find, because modern agricultural activities represented by rice farming have profoundly changed the surface morphology, and frequent human activities and abundant precipitation have significantly changed the surface morphology. Therefore, in previous studies, there are few studies using CORONA satellite images in this region.

In fact, due to abundant precipitation and numerous rivers and lakes in this region, most villages from prehistoric to bronze age and even modern times are distributed on the soil mounds. This settlement location method can well avoid flood threats in the rainy season. These mounds, which are 3–5 m higher than the surrounding ground surface, are the focus of archaeologists when conducting field investigations. In the CORONA satellite image, such square or circular mounds that are higher than the surrounding ground surface can be clearly identified. However, due to the limited resolution of the

CORONA image itself (2–8 m), when we need to observe the detailed features of the mound, the information provided by such satellite images is somewhat insufficient. UAV photogrammetry technology can make up for the shortage of satellite images, and the DOM resolution generated by it can reach 2–5 cm. At the same time, with the help of the software ArcGIS, DEM can present the mound structure in detail so that we can find some information that was ignored in the past, such as the city wall distributed on the edge of the mound or the ditches that have been silted up now. Therefore, the combination of CORONA satellite and UAV photogrammetry has proved to be an effective way to carry out an archaeological survey in the Yangtze River basin.

In the above research cases, CORONA and UAV images show their respective advantages. CORONA images provide precious images before human beings significantly changed the surface morphology half a century ago, and UAVs can quickly obtain high-resolution surface elevation information and orthophoto images. It can be said that UAVs are the most promising tool in the field of remote-sensing archaeology [18–20]. However, UAVs also have some disadvantages, such as limited battery capacity, which results in a single flight time of about half an hour. Part of the airspace is restricted by flight, and the altitude and range of flight are restricted. These factors restrict the application of UAVs in archaeological research. However, we still have reason to believe that UAV technology will make greater progress in the future, which will serve the field archaeological work better.

## 6. Conclusions

The combination of UAV photogrammetry and CORONA satellite image can improve the research ability of archaeologists in many aspects, including reconstruction of settlement layout, detection of unknown archaeological features and sites, research of settlement location and construction methods, and regional landscape dynamics. Therefore, the application of multisource remote sensing images, including UAV, CORONA images, hyperspectral images, thermal infrared images, etc., in field archaeological work is the trend and direction of remote sensing archaeological development in the future. How to integrate different remote sensing images and technical means in complex geographical environments and climatic conditions is a question that archaeologists should constantly consider and learn.

**Funding:** This work was supported by the National Key R&D Program of China (Project No. 2022YFF 0903600), and National Social Science Foundation of China (Project No. 22CKG013), and the China Postdoctoral Science Foundation (Project No. 2020M682463).

**Institutional Review Board Statement:** Not applicable.

**Informed Consent Statement:** Not applicable.

**Data Availability Statement:** Restrictions apply to the available of these data. Data was obtained from Wuhan University and are available from the corresponding author with the permission of Wuhan University.

**Acknowledgments:** We are very grateful to the following individuals for their contributions to the project: Changping Zhang and Haichao Song. We are also grateful to the many graduate students who have joined us in the field or otherwise contributed to this work: Zheng Gangfan, Liao Hang, Xu Shen, and Lu Jindong. We would also like to thank the following institutions for their generous support of our fieldwork: The School of History at Wuhan University, Archaeological Institute of Hubei Province, Panlongcheng Site Museum, Huangmei county Museum, Dawu County Museum.

**Conflicts of Interest:** The authors declare no conflict of interest.

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
