# Peer review of "Detecting the Bronze Age Sites by Using CORONA Satellite Photography and UAV Photogrammetry: A Case Study from the Middle of Yangtze River, China"

_land, doi:10.3390/land12030685_

Round 1

Reviewer 1 Report

Dear author,

I went through, with much interest, your manuscript, and I can see that it has potential for publication. However, I have the following recommendations:

1-    In the title, you stated, ‘Detecting the bronze age sites…’; however, this means that you have identified some new archaeological sites that have not been discovered before. If this was the case, you should show us how many new sites you found using CORONA satellite images. Then, you can either list the newly discovered sites or display them on a separate map.

2-    In figure 1, it is better to add a legend to the map and also if you point to some modern cities or towns on the map. For example, you can add the word ‘China’ to the topographic map ( in the top left).

3-    In figure 1, I can see 12 sites, are they been discovered in this study? How many of them do you identify some new archaeological features?

4-     Unfortunately, you gave us only two examples of that ( figure 2 and 3). You should provide us with more diverse examples.  

5-    In figure 3, it seems you haven’t used CORONA for the reconstruction; why not? If you did, show us, please?   

6-    In the discussion section, you mentioned the arid and semi-arid regions but haven’t cited any publications. See those examples, and you can see how CORONA satellite images were also used to reconstruct archaeological sites in wetland areas. So, in my opinion,  CORONA is still helpful in your area too. However you can argue against that, but you should give more examples and details.  

-Hritz, C. (2010). Tracing settlement patterns and channel systems in southern Mesopotamia using remote sensing. Journal of Field Archaeology35(2), 184-203.     

-Jotheri, J.; Feadha, M.; Al-Janabi, J.; Alabdan, R. Landscape Archaeology of Southern Mesopotamia: Identifying Features in the Dried Marshes. Sustainability 202214, 10961. https://doi.org/10.3390/su141710961

-Pournelle, J. R. (2007). KLM to CORONA: A bird’s-eye view of cultural ecology and early Mesopotamian urbanization. Settlement and society: essays dedicated to Robert McCormick Adams, 29-62.

7-    You haven’t mentioned using other satellite images or historic maps to reconstruct or identify archaeological sites. For example, declassified Hexagone satellite images or the historic Google Earth etc.      

 All the best

Author Response

  • In the title, you stated, ‘Detecting the bronze age sites…’; however, this means that you have identified some new archaeological sites that have not been discovered before. If this was the case, you should show us how many new sites you found using CORONA satellite images. Then, you can either list the newly discovered sites or display them on a separate map.

Response:What I want to express in this paper is discovery of unknown features in archaeological sites( such as walls and moats) by using drones and satellite imagery and now I have added a table in the paper that records the details of these sites.

  • In figure 1, it is better to add a legend to the map and also if you point to some modern cities or towns on the map. For example, you can add the word ‘China’ to the topographic map ( in the top left).

Response:I have modified Figure 1 and added latitude and longitude to the periphery of the frame.

  • In figure 1, I can see 12 sites, are they been discovered in this study? How many of them do you identify some new archaeological features?

Response:All the 12 sites have been described in published archaeological reports. But the features of these sites and construction methods have not been recorded, and this information has been discovered by use with the help of drones and fieldwork.

  • Unfortunately, you gave us only two examples of that ( figure 2 and 3). You should provide us with more diverse examples.  

Response:OK,I added a table in the article to record the details of more sites.

  • In figure 3, it seems you haven’t used CORONA for the reconstruction; why not? If you did, show us, please?   

Response:I used CORONA for the reconstruction(Figure3,2).We first discovered the walls along the edge of the site through CORONA satellite imagery, and then we used drones to survey this site(Yishengsi).

  • In the discussion section, you mentioned the arid and semi-arid regions but haven’t cited any publications. See those examples, and you can see how CORONA satellite images were also used to reconstruct archaeological sites in wetland areas. So, in my opinion,  CORONA is still helpful in your area too. However you can argue against that, but you should give more examples and details.  

Response:I agree with you and have addes more references in this paper.

  • You haven’t mentioned using other satellite images or historic maps to reconstruct or identify archaeological sites. For example, declassified Hexagone satellite images or the historic Google Earth etc.      

Response:In this case study, we did not use other satellite imagery or historic maps,so I did not mention them.

Reviewer 2 Report

The author presents a reanalysis of CORONA imagery in the context of an archaeological survey of the Middle Yangtze River, alongside additional data collection and generation of DSM and orthoimagery with a UAV. The author correctly asserts that several remote sensing datasets can complement each other to reconstruct site layout, urban design, and social structure in archaeological contexts. The research presented here is sound and significant, but the manuscript needs extensive editing and restructuring to ground the research theoretically and to frame the importance of the research.

The manuscript clearly presents the value and results of the research, but the author provides insufficient structure from the outset. The paper in its current state lacks an overall argument. In my reading of the manuscript, the clearest argument seems to be that only through integrating historical, legacy datasets with modern techniques can site form and layout properly be understood, leading to a reinterpretation of this region's social structure. However, the paper provides no context of what previous researchers have written about this region's social structure. Readers with minimal understanding of the culture history will not be able to interpret how this study changes our understanding of this region's social structure.

The author should therefore highlight this argument (if this is indeed the argument) from the outset, followed by a brief discussion of culture history and previous work. Once structured in this manner, readers will be more able to acknowledge the significance of this research.

The author should also revise the paper carefully for typos and errors, although the writing in general is fairly clear.

On line 41, the author discusses obvious limitations in using CORONA in the context of landscape archaeology. These limitations are not obvious and should be clearly described.

Around line 226 the author mentions social structure, which again should be brought up earlier to frame the paper.

Line 236 mentions construction method, which is unclear -- at first I thought this referred to construction materials but the author seems to be discussing site layout and design.

Line 239 is worded in a very confusing manner -- 90% of the site is 10,000 square meters?

Line 243 again seems to be a misuse of wording, "landscape of the site" is used when I think the author is referring to site typology.

Line 278 seems to be the thesis of the paper, but it arrives very late. Complex construction methods correlate with higher social levels (I do not necessarily agree with this statement, but if this is the author's argument, it should be clearly stated from the outset and defended).

In line 319, the author claims that UAVs are the most promising tool in archaeological remote sensing. This statement requires further discussion and citations.

Overall, I think this is a significant study worth publishing, but the manuscript needs to be restructured and given a more robust theoretical grounding to do justice to the author's impressive research.

Author Response

On line 41, the author discusses obvious limitations in using CORONA in the context of landscape archaeology. These limitations are not obvious and should be clearly described.

Response: Ok, I have discussed more clearly about the limitations of CORONA images now.

Around line 226 the author mentions social structure, which again should be brought up earlier to frame the paper.

Response: Ok,I discussed social structure at the beginning of my article.

Line 236 mentions construction method, which is unclear -- at first I thought this referred to construction materials but the author seems to be discussing site layout and design.

Response: “construction method” means the layout and structure of these sites in this paper.

Line 239 is worded in a very confusing manner -- 90% of the site is 10,000 square meters?

Response: I have added a table (Table1)to the article, which indicates the coordinates and area of each site. And I changed “90%” to “most” in this paper.

Line 243 again seems to be a misuse of wording, "landscape of the site" is used when I think the author is referring to site typology.

Response: Ok, I replaced “landscape” with ”type”.

Line 278 seems to be the thesis of the paper, but it arrives very late. Complex construction methods correlate with higher social levels (I do not necessarily agree with this statement, but if this is the author's argument, it should be clearly stated from the outset and defended).

Response: Ok, I've added more statements and explanations here.

In line 319, the author claims that UAVs are the most promising tool in archaeological remote sensing. This statement requires further discussion and citations.

Response:I've added more references here.

Reviewer 3 Report

See Comments sent to the editor.

Author Response

Line 17 – reconstructing, not reconstruction

Line 25 – remove the word ‘spy’ – not necessary/needed

Line 32 – in Egypt, not at Egypt

Line 35 – extra period after ‘sites’

Response:I have fixed the above typos and grammar errors.

Line 54 – what do you mean by ‘mounds’? Hills?

Response:In this paper,‘mounds’ means small earth hills 3-5 meters above the ground.

Line 63 – Use another term than ‘infrastructure construction’ especially if not for urban

Development

Response:I this‘infrastructure construction’can express exactly what I mean.What I want to express here is that large-scale infrastructure construction was not carried out before the 1970s.

Line 78 – Need calendar dates for Bronze Age; also need to capitalize it consistently in the

manuscript.

Response:The calendar dates of Bronze age in China is 1600BC-256BC,and I have added this data to this manuscript.

Line 103 – (wall, moat)

Line 108 – download of CORONA images…

Line 110 – Finally, the…

Line 115 – Survey

Line 152 – acquisitioned, not acquisition

Response:I have fixed the above typos and grammar errors.

Line 164 – What do you mean by drilling/ testing? – see also lines 185 and 225.

Response:Archaeological drilling means using a drilling tool to carry out exploration work to judge the distribution of underground archaeological remains by observing the inclusions in the soil and the color of the soil.

Line 166 – detecting, not detection

Line 233- Is that 30 sites plus Yishengsi and Lvwangcheng or 32 sites including those two?

Response:Yes,that 30 sites already plus YIshengsi and Lvwangcheng.

Line 286 – do you mean spatial instead of space?

Response:Yes, I means spatial.

Lines 293 and 301 – What is the difference between prehistoric and Bronze Age?

Response:Prehistoric means Paleolithic and Neolithic,the calendar dates of prehistoric age is before 1700BC. And the calendar dates of Bronze age is 1600BC-256BC.

Line 307 – extra spaces between the comma

Based on the examples above there are a number of small grammatical issues but they do not

detract from the quality of the manuscript. Overall, the author does a goof job of explaining the

differences between the old and new technology. I’d like to see the results of the next phase of

fieldwork that would go beyond survey and involve testing at many of these sites to see if

ground-truthing the imagery pays off. One thing that I would like to see in this manuscript is a

table that provides information on all 32(?) sites. The table can be set up with columns that show

the CORONA and UAV data (coordinates but also length, width, height of each site) along with

the different settlement types associated with each. It would be nice to have all of this data

included in a table. The methodology is clear and the figures are excellent. The bibliography is

current and relevant. I think the manuscript should be published once the typos are fixed and a

table added

Response:

Table1: Sites coordinates and layout information

Site

Coordinate

Elevation

Shape

Area(㎡)

Dunzifan

114.3253E,31.659817N

163.5

Elliptic

1,751

Sigudun

114.416048E,31.423182N

64.9

Circle

16,726

Leijiazui

114.420243E,31.324574N

45.5

Circle

2,815

Fumadun

114.503509E,31.550313N

82.7

Square

5,623

Yishengsi

115.8733177E,30.03612684N

12.9

Elliptic

100,527

Shaishutai

113.6375892E,31.19509047N

41.2

Square

7,956

Zhaishang

115.0394661E,30.62190842N

36.2

Elliptic

20,262

Zengjiadun

114.6564424E,30.89109422N

50.9

Rectangular

9,532

Dadun

114.6010923E,30.8495437N

34.4

Elliptic

9,332

Zhaidun

114.6302426E,30.89518199N

34.5

Elliptic

2,289

Xiaowangjiashan

113.7747788E,31.00284662N

24.1

Elliptic

14,245

Xiayaozui

114.8081374E,30.69486987N

27.6

Rectangular

50,223

Lvwangcheng

114.4766255E,31.50678997V

75.2

Elliptic

42,729

Round 2

Reviewer 2 Report

I reviewed an earlier version of this manuscript. The authors have clarified most of my concerns, but the paper still lacks a clear, overarching argument. Adding an argument can be done easily by including a paragraph at the end of the introduction, briefly summarizing the structure of the paper and crafting a thesis statement. Something along the lines of "our research shows that integrating legacy datasets with new UAV technologies has provided new insight into archaeological site design and social structure along the middle Yangtze River."

Author Response

Dear reviewer, I have added a paragraph to the outline of the article as you suggested the idea of this project. Exploration of the role of UAV photogrammetry in field archaeology has also been added. Thank you for reviewing the manuscript again.
